# The role of surface adhesion on the macroscopic wrinkling of biofilms

Steffen Geisel[1], Eleonora Secchi[2]*, Jan Vermant[1]*

[1]Laboratory for Soft Materials, Department of Materials, ETH Zurich, Zurich, Switzerland; [2]Department of Civil, Environmental and Geomatic Engineering, ETH Zurich, Zurich, Switzerland

**Abstract** Biofilms, bacterial communities of cells encased by a self-produced matrix, exhibit a variety of three-dimensional structures. Specifically, channel networks formed within the bulk of the biofilm have been identified to play an important role in the colonies' viability by promoting the transport of nutrients and chemicals. Here, we study channel formation and focus on the role of the adhesion of the biofilm matrix to the substrate in *Pseudomonas aeruginosa* biofilms grown under constant flow in microfluidic channels. We perform phase contrast and confocal laser scanning microscopy to examine the development of the biofilm structure as a function of the substrates' surface energy. The formation of the wrinkles and folds is triggered by a mechanical buckling instability, controlled by biofilm growth rate and the film's adhesion to the substrate. The three-dimensional folding gives rise to hollow channels that rapidly increase the effective volume occupied by the biofilm and facilitate bacterial movement inside them. The experiments and analysis on mechanical instabilities for the relevant case of a bacterial biofilm grown during flow enable us to predict and control the biofilm morphology.

*For correspondence:
esecchi@ethz.ch (ES);
jan.vermant@mat.ethz.ch (JV)

**Competing interest:** The authors declare that no competing interests exist.

## Editor's evaluation

The wrinkling of growing biofilms is considered in this paper experimentally in a clever set of experiments in a microfluidic setup that reveals aspects of the onset of the wrinkling instability and the formation of hollow channels within which bacteria move. Variations in the adhesive properties of the underlying surface are shown to affect the instability.

## Introduction

Bacteria predominantly exist in biofilms, surface-attached aggregates of cells (*Nadell et al., 2017*; *Flemming et al., 2016*; *Flemming and Wuertz, 2019*). In biofilms, the cells are enclosed in auto-produced, strongly hydrated extracellular polymeric substances (EPS), which form the extracellular matrix. The EPS consists of polysaccharides, the most abundant component, proteins, nucleic acids, and lipids (*Lasa, 2006*; *Frølund et al., 1996*; *Flemming and Wingender, 2010*). The matrix plays different roles: its viscoelastic nature provides mechanical stability to the biofilm, while its physical chemistry is responsible for the adhesion to the surface and internal cohesion (*Costerton et al., 1987*; *Hall-Stoodley et al., 2004*). Additionally, not only mechanical and chemical but also the matrix's structural properties contribute to the exceptional viability of the bacterial community in the biofilm lifestyle (*Epstein et al., 2011*; *Okegbe et al., 2014*; *Madsen et al., 2015*). However, the mechanistic understanding of how environmental conditions and the characteristics of the surfaces on which they grow affect the biofilm structure is still limited.

Bacterial biofilms are found in a vast range of environments and applications, ranging from bioremediation (*Ghosh et al., 2019*) to biomedical (*Badal et al., 2020*; *Bixler and Bhushan, 2012*) and

industrial fouling (*Schultz et al., 2011*). In most settings, the biofilm forms on a solid surface while being exposed to fluid flow. Hydrodynamic conditions control mass transfer, which in turn controls the transport of nutrients, metabolic products, and signal molecules (*Purevdorj et al., 2002*; *Krsmanovic et al., 2021*; *Conrad and Poling-Skutvik, 2018*). Fluid flow also exerts drag forces on the biofilm and shapes its structure (*Stoodley et al., 1998*; *Stoodley et al., 1999*; *Pearce et al., 2019*; *Hartmann et al., 2019*). Under strong flows, bacteria often form biofilm streamers in the shape of long, filamentous structures; while, under weak flow conditions, some bacteria form surface-attached colonies with ripple-like structures (*Rusconi et al., 2011*; *Rusconi et al., 2010*; *Purevdorj et al., 2002*; *Drescher et al., 2013*). However, it is unclear what mechanisms govern the structure evolution under flow, which is most often present. Therefore, understanding biofilm morphogenesis under hydrodynamically relevant conditions is of crucial importance both from the biological and engineering standpoint.

Some biofilms exhibit three-dimensional morphologies characterized by the presence of folds and wrinkles that have been proposed to improve the viability of the biofilm due to improved uptake and transport of oxygen and nutrients (*Wilking et al., 2012*; *Kempes et al., 2014*). Many experimental studies focused on static biofilm-agar systems to characterize the mechanical contributions to the formation of these 3D structures (*Asally et al., 2012*; *Yan et al., 2019*). However, a biofilm grown on agar induces additional complexities, as biological and mechanical contributions are tightly interconnected. Additionally, it may not be as relevant for biofilms occurring in industrial or natural environments where fluid flow and solid substrates are often present. Biofilms grown on agar are characterized by substantial heterogeneity in nutrient availability, generated by the diffusive nature of nutrient transport within the agar, which leads to differences in growth rates and subsequent mechanical stresses across the biofilm (*Stewart, 2003*; *Wilking et al., 2011*). Moreover, the complexities associated with the motion near the contact line complicate matters further. Theoretical and experimental studies found that, in particular, an anisotropic growth may be the driving force for folding in colonies with moderate adhesion to the substrates (*Ben Amar and Wu, 2014*; *Espeso et al., 2015*; *Fei et al., 2020*). Additionally, how the biofilm colony can spread across the agar plate as the biofilm is governed by a complex interplay with bulk and interfacial (Marangoni) stresses (*Verstraeten et al., 2008*; *Seminara et al., 2012*; *Srinivasan et al., 2019*). Growth gradients and colony spreading are relatively poorly understood processes that involve both biological as well as mechanical effects and hence make biofilm on agar not ideal as a model system to isolate the effects of mechanical contributions alone (*Zhang et al., 2016b*; *Fauvart et al., 2012*; *Gingichashvili et al., 2020*). In this study, we aim to deconvolute the interplay between mechanical forces and biological contributions, such as inhomogeneous growth of the microorganisms, by laterally confining the biofilm to control its spreading and provide it with a controlled, homogeneous, and constant supply of nutrients in a microfluidic channel. The goal is to advance the understanding of simple mechanical contributions to folding and wrinkling of biofilms further.

The effects of mechanical stresses on the formation of three-dimensional morphologies are well understood in several eukaryotic systems, including ripple-shaped leaves or the fingerprints of humans. Often these structures are developed due to bonded layers of biomaterial and cells that grow at different rates (*Liang and Mahadevan, 2011*; *Kücken and Newell, 2004*; *Savin et al., 2011*). Similar morphologies with an origin in mechanical instabilities have been investigated in thin-film studies, when elastic films are attached to a stiff substrate and compressive stresses are induced chemically or thermally (*Hutchinson et al., 1992*; *Chung et al., 2011*; *Cerda and Mahadevan, 2003*). Common characteristics of these biological and artificial examples are adhesion between the layers and a mechanical strain mismatch. The consequently induced compressive stress leads to various morphologies such as wrinkles, folds, or delaminated blisters (*Wang and Zhao, 2015*). Although the structures found in bacterial biofilms show many qualitative similarities, only recently the links between folds in biofilms and mechanical instabilities have been investigated. Recent studies found that the adhesive strength and friction between biofilm and substrate might play a role in virulence as well as the structural evolution of the biofilm (*Fei et al., 2020*; *Cont et al., 2020*). However, many experimental studies use agar as a substrate where adhesion appears to be spatially and temporally heterogeneous (*Gingichashvili et al., 2020*). Therefore, systematic investigations of the interplay between adhesive strength and fold formation are needed to better understand the mechanical instabilities that govern biofilm morphology.

In this work, we report on the structural evolution of confined biofilms grown under well-controlled flow conditions. We investigated the basic mechanism of biofilm folding and wrinkling under well-defined conditions relevant to environmental, industrial, and medical settings. We show for the first time how wrinkling of a *P. aeruginosa* PAO1 biofilm creates hollow channels occupied by motile bacteria. Our results indicate that for a laterally confined biofilm, growth on a solid substrate induces compressive stresses that are the key driving force for buckling-delamination that governs the formation of channel networks. The process of buckling-delamination is expected to depend on the material properties of the biofilm, growth-induced compressive stresses, and the adhesive strength between the biofilm and the solid substrate. Experimentally, we can readily control the biofilm adhesion to the substrate. Consequently, the biofilm morphology can be spatially controlled and patterned, giving unprecedented control over the macroscale structure and the average thickness of the biofilm.

## Results

### Wrinkle formation at the solid-liquid interface

*P. aeruginosa* biofilms grown on a solid surface are exposed to controlled flow in a microfluidic device. They form wrinkles that span the entire biofilm. The microfluidic device consists of a simple rectangular channel, made of polydimethylsiloxane (PDMS) bonded onto a glass slide and mounted onto an inverted microscope. The channel is $500\,\mu m$ wide, $100\,\mu m$ high, and $1.5\,cm$ long (*Figure 1a*). The microfluidic channel was filled with a PAO1 bacterial suspension at $OD_{600} = 0.2$ and left at rest for 1 hr before the flow of fresh culture medium was started. We use a positive displacement syringe pump to control the flow of the nutrient solution at an average flow rate of $0.3\,ml\,h^{-1}$ resulting in an average velocity of $1.7\,mm\,s^{-1}$. This leads to an average wall shear stress of $0.1\,Pa$, several orders of magnitude lower than the elastic shear modulus of *P. aeruginosa* biofilms reported in the literature (*Lieleg et al., 2011*; *Secchi et al., 2022*). Bacterial cells exposed to the flow of nutrients first grow as a uniform layer of increasing thickness on the PDMS. No significant biofilm formation is observed on the glass within the timeframe of our experiments. As the biofilm grown on the PDMS reaches a thickness of $10\,\mu m$ to $20\,\mu m$ after 48–72 hr, the biofilm develops a pattern of folds and wrinkles (*Figure 1—video 1*). The pattern is qualitatively similar to the structures observed in previous studies, where the biofilm was grown under static conditions on agar plates (*Kempes et al., 2014*; *Yan et al., 2019*). The initial wrinkle formation starts with small wrinkles that start to appear throughout the biofilm and are visible in the phase-contrast images of *Figure 1b* panel II. The wrinkles typically have a size of $10\,\mu m$ to $30\,\mu m$ in diameter when they can first be identified. The wrinkles evolve over several hours into an interconnected pattern visible as dark lines in the phase-contrast time-lapse images of *Figure 1b*. In the final stage of their development, the individual wrinkles have a width of $20\,\mu m$ to $30\,\mu m$ and the connected network spans the whole biofilm. It can be noted that the presence of wrinkles and voids increases the effective average thickness of the biofilm. The wrinkle formation process, the wrinkle size, and the effective thickness increase was qualitatively reproducible over more than 20 experiments, whereas the number of wrinkles is subject to some biological variability.

The temporal evolution of the structure can be divided into three distinct stages. We define the stages by quantifying the number of individual isolated wrinkles $N$ and the length of the longest connected wrinkle $L$. The low magnification phase-contrast images were binarized and subsequently skeletonized (for details, see section Skeletonization of channel networks) to extract the desired parameters, namely the number of isolated wrinkles $N$ and the length of the longest wrinkle $L$ (*Figure 1b and c*). The first stage starts shortly before the first wrinkles appear, which is 49 hr after the start of the nutrient flow, and lasts 6.5 hr. The first stage is characterized by a substantial increase in the number of isolated wrinkles, while the length of the longest wrinkle remains small ($L < 0.65\,mm$). In the optical observation, at this stage, many small and isolated wrinkles develop evenly throughout the biofilm. In the second phase, which lasts approximately 3.5 hr, the number of isolated wrinkles decreases because they start to merge and form longer, interconnected paths. This results in a few remaining wrinkles with a considerable length in the order of $8\,mm$ that form a highly connected network throughout the biofilm. Finally, in the third phase, the biofilm structure reaches a steady state where the longest wrinkle does not grow in length anymore and the number of unconnected wrinkles stays consistently low. This final stage has been observed to last at least 5 hr, while the whole process

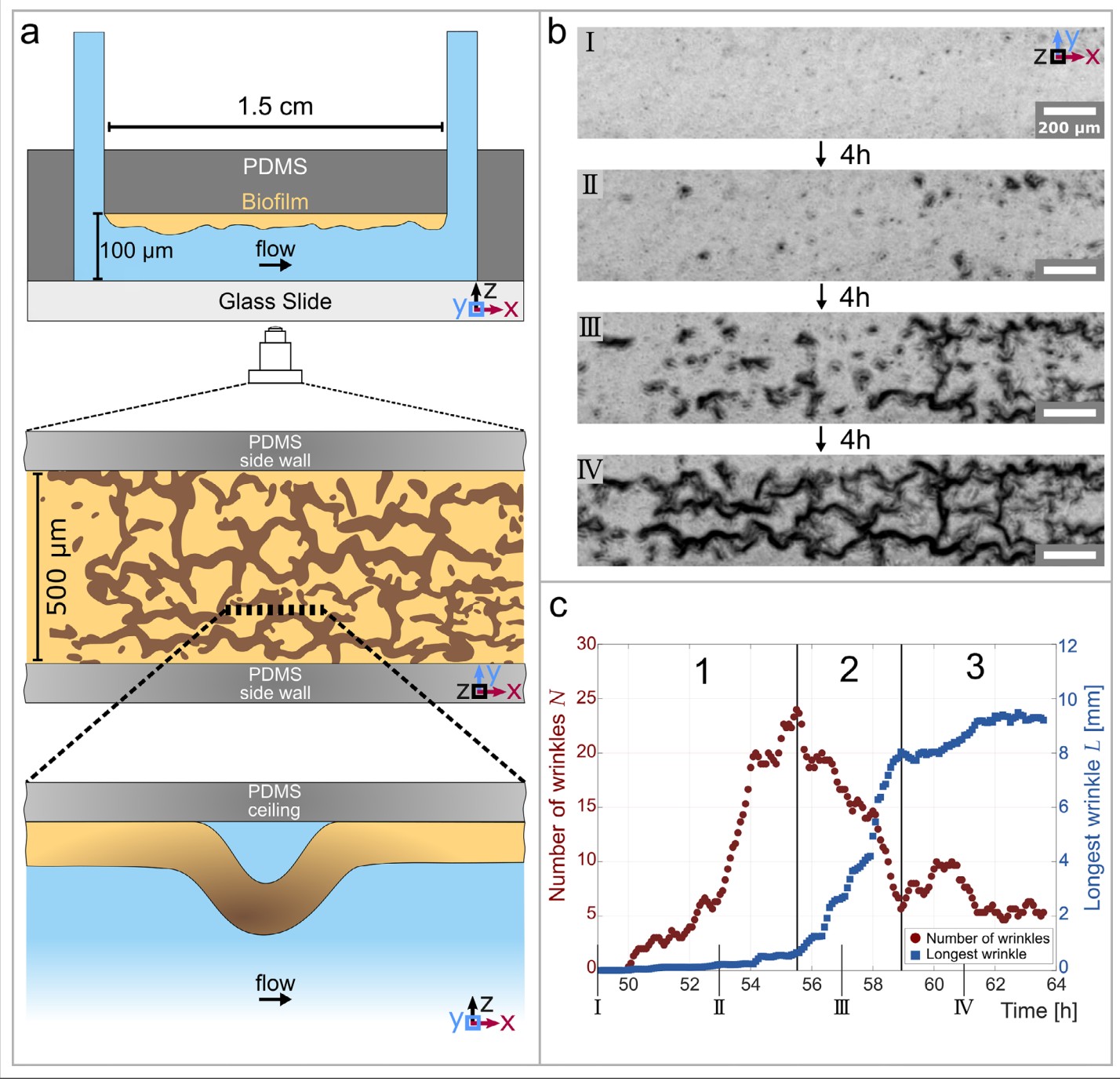

**Figure 1.** Temporal and structural evolution of wrinkles in *P. aeruginosa* PAO1 biofilms grown in flow. (**a**) Schematic representations of the microfluidic device, the wrinkle network in the biofilm and a side view of a single wrinkle. (**b**) Time evolution of the wrinkled structure in the biofilm. Images were taken in phase contrast. (**c**) Number of individual wrinkles, $N$ (red) and the length of the longest connected wrinkle $L$ (in *mm*, blue). The evolution of the wrinkled biofilm can be divided into three distinct stages. Many small, isolated wrinkles appear in the first stage. The wrinkles connect to form a network in the second stage. In the third stage, the biofilm has reached a steady state.

The online version of this article includes the following video for figure 1:

**Figure 1—video 1.** Timelapse video of the wrinkle formation from a flat biofilm to the completely develop wrinkle network.

https://elifesciences.org/articles/76027/figures#fig1video1

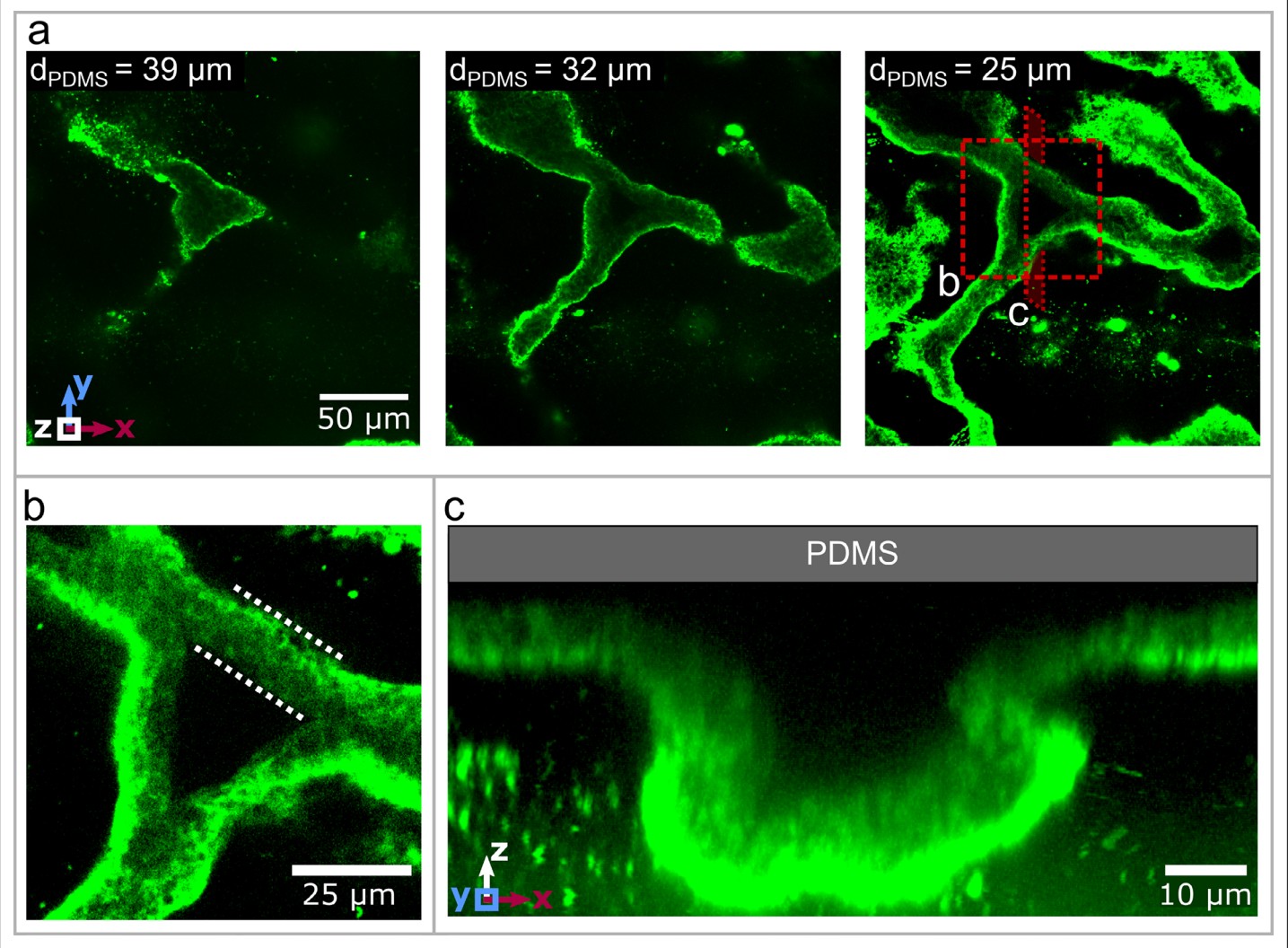

**Figure 2.** Three dimensional structure of the biofilm wrinkles. (**a**) Laser-scanning confocal microscopy images of a biofilm that developed a channel network, stained with a GFP-labelled Concanavalin A lectin stain. The three images show slices in the x-y-plane, starting 39 $\mu m$ away from the PDMS surface. The second image is taken 32 $\mu m$ and the third image 25 $\mu m$ away from the PDMS substrate. (**b**) Close up of the biofilm channel shown in panel (**a**). The white, dotted lines indicate the walls of the biofilm channel. (**c**) Cross-section and close up of a channel along the cutting plane indicated in panel (**a**).

of biofilm wrinkling proceeds over 9–10 hr once the first wrinkles appear and until a steady state is reached.

## Wrinkles create three-dimensional channels

Detailed imaging of fluorescently labeled biofilm with confocal laser scanning microscopy reveals the three-dimensional topology of the wrinkles. We stain the polysaccharide component of the biofilm matrix with GFP-fluorescent Concanavalin A and use confocal microscopy to image biofilm wrinkles in the x-y-plane at different distances from the PDMS substrate (*Figure 2a*). The first image shows the very top of a wrinkle, 39 $\mu m$ away from the PDMS. As we move closer to the biofilm's base, the extent of the network becomes visible, with connected wrinkles reaching across the whole field of view of 200 $\mu m$. To visualise the topography of the biofilm, consider a simple piece of fabric on a solid substrate. If the fabric gets pushed together, it locally separates from the substrate to form a three-dimensional pattern with folds and wrinkles that resemble the biofilm.

The analogy of a wrinkled fabric can be extended to the internal structure of the biofilm wrinkles. The wrinkles consist of hollow channels that detach from the substrate during their formation. A

close-up image of a wrinkle $25\,\mu m$ away from the PDMS (*Figure 2b*) allows us to define the walls of the wrinkle, which are rich in biofilm matrix according to the strong fluorescent signal. In contrast, the center of the wrinkle does not show a fluorescent signal and is therefore devoid of any biofilm matrix. This result demonstrates that the wrinkles create hollow channels with walls made out of biofilm matrix, and in the remaining course of this paper, we will refer to this as a channel network. In order to form a channel network, the biofilm locally needs to detach and buckle away from the substrate. A vertical slice through a confocal volume along the x-z-plane (*Figure 2c*) shows that the channel height is substantially greater than the thickness of the original biofilm layer. Furthermore, no biofilm matrix was detected on the PDMS substrate at the location of the channel. This indicates that the biofilm fully detaches from the substrate, similar to our analogy where the fabric separates from the substrate to form a pattern of wrinkles and folds. This delamination between the biofilm and the PDMS substrate allows us to identify buckling-delamination as the underlying mechanism driving the formation of a channel network throughout the biofilm.

## Buckling-Delamination as the driving force for channel formation

The growth of the biofilm in a confined environment acts as the driving force for the buckling instability, which leads to the formation of the channel network. In our experiments, we control the nutrient availability – and therefore the growth rate of the biofilm – in a biofilm on a solid, planar surface. As reported in *Figure 3a*, a biofilm is first grown under standard experimental conditions with a constant flow of nutrients. Seven hours after the appearance of the first channels, the nutrient solution is replaced with a salt solution of equal salinity but devoid of any nutrients. After 18 hr without any nutrient supply, the salt solution is replaced again with the nutrient-rich solution the biofilm was initially grown in and supplied nutrients for an additional 24 hr. The evolution of the number of isolated channels, $N$, as a function of time and nutrient availability (*Figure 3a*) demonstrates that the steep increase in $N$ is abruptly interrupted when the biofilm is no longer supplied with nutrients. An increase or change in $N$ only occurs when nutrients are present. The channel formation restarts as the nutrient solution is reintroduced in the microfluidic channel and continues, as shown in *Figure 3a* (right panel). The switch from a nutrient-rich to a nutrient-depleted solution inhibits biofilm growth reproducibly without changing any environmental conditions such as flow velocity, temperature, and salinity. Therefore, we can unambiguously identify biofilm growth as the key driving force for the formation of a channel network.

The structural analysis of the channels and identifying the biofilm growth as the driving force controlling channel formation lead us to conclude that a buckling-delamination process governs channel formation. *Velankar et al., 2012* analyzed the swelling-induced buckling of a thin elastic film loosely bound to a stiff substrate. The same approach can be applied to the formation of a biofilm as schematically depicted in *Figure 3b*. In the initial stage, *Figure 3b–I*, the surface is populated by bacteria that grow and form a biofilm. The growth of the biofilm within the constrained space of a microfluidic channel results in compressive stresses $\sigma$, which are presumed to be uniform and equibiaxial (*Figure 3b–II*). The biofilm is assumed to have isotropic mechanical properties with Young's modulus $E_f$, Poisson's ratio $\nu_f$ and thickness $h$. We now consider a circular, delaminated blister with radius $R$, where the adhesion between the film and the substrate is minimal (*Figure 3b–III*). In the unbuckled state, the energy release rate of the interface crack is zero and the blister will not grow. Only when the film buckles away from the substrate, the crack driving force will be nonzero. The critical stress when the film buckles away from the substrate is given by *Hutchinson et al., 1992* as

$$\sigma_c = 1.2235 \frac{E_f}{1-\nu_f^2} \left(\frac{h}{R}\right)^2. \tag{1}$$

The biofilm will buckle away from the substrate for stresses greater than $\sigma_c$.

Analysis of the microscopy images described in section wrinkle formation at the solid-liquid interface showed that the initial delaminated blisters are $5\,\mu m$ to $15\,\mu m$ in radius and the height of the homogeneous, non-wrinkled biofilm is $10\,\mu m$ to $20\,\mu m$. In order to estimate the magnitude of the critical compressive stress $\sigma_c$ needed to delaminate such an elastic film, we choose $R = 10\,\mu m$ and $h = 15\,\mu m$. The elastic shear modulus of *P. aeruginosa* biofilms has been found to be of the order of $\sim 1000\,Pa$ and, similarly to previous studies, we assume $\nu_f$ to be 0.45 (*Kundukad et al., 2016*; *Lieleg et al., 2011*). These values give a critical stress of $\sigma_c \approx 3500\,Pa$ needed for the biofilm to buckle

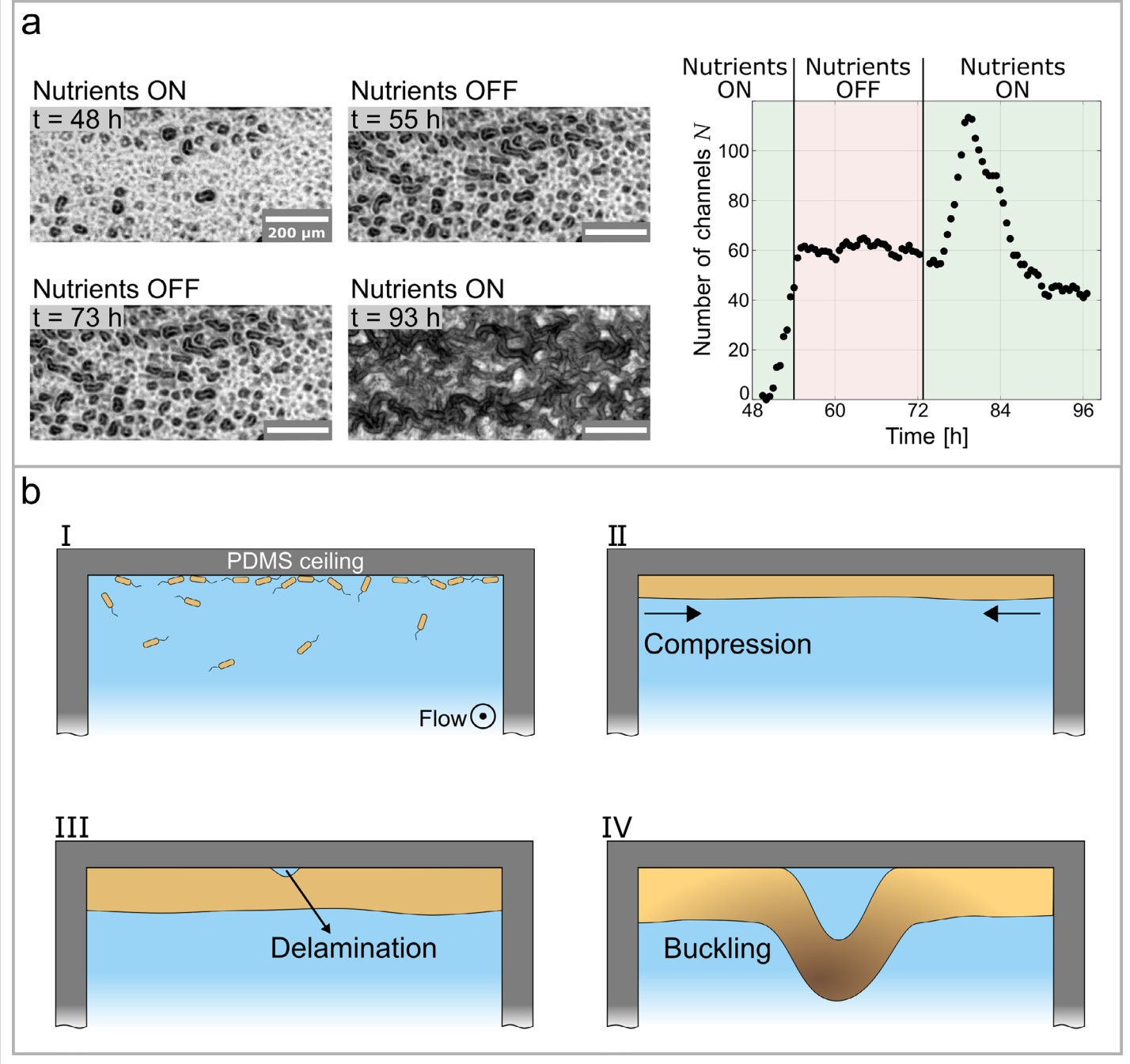

**Figure 3.** Growth controls the formation of channels through buckling-delamination. (**a**) Phase-contrast images of the experiment conducted to investigate the role of biofilm growth in channel formation. At $t = 55\,\text{h}$ the nutrient solution is replaced with a nutrient-depleted salt solution to stop growth. At $t = 73\,\text{h}$ the salt solution is replaced with the original nutrient solution. The graph shows the number of channels, $N$, as a function of time and nutrient availability. (**b**) Schematic representation of the buckling-delamination mechanism during channel formation in *P. aeruginosa* biofilm in a microfluidic device.

away from the substrate. Previous experimental studies have reported that *P. aeruginosa* biofilms are indeed capable of exerting stresses in the kPa-range during growth (***Chew et al., 2016***).

Once buckled, the criterion for subsequent growth of the delaminated blister is dependent on the driving force for crack propagation, the energy release rate $G$. The elastic energy per unit area stored in the unbuckled film is equal to $G_0 = (1 - \nu_f)h\sigma^2/E_f$. The ratio $G/G_0$, so the crack driving force $G$ normalized by the elastic energy stored in the unbuckled film $G_0$, depends only on the compressive

stress $\sigma$, the critical compressive stress $\sigma_c$ and the Poisson's ratio $\nu_f$ of the material in the film and can be expressed as

$$\frac{G}{G_0} = c_2 \left[ 1 - \left( \frac{\sigma_c}{\sigma} \right)^2 \right] \tag{2}$$

where $c_2 = [1 + 0.9021(1 - \nu_f)]^{-1}$. So the energy release rate $G$, which drives crack propagation, increases monotonically with $\sigma/\sigma_c$, approaching $G_0$. Therefore, the crack propagation's driving force increases with the growth-induced stress $\sigma$. This theoretical assessment shows that the growth-induced compressive stress and the observed size of the initial delaminated blisters are sufficient to induce and drive buckling-delamination of biofilms. Additionally, delamination and the advance of the interface crack are dependent on the interface toughness $\Gamma$, defined as the resistance to the propagation of an interface crack. This interface toughness $\Gamma$ depends on the deformation mode, which remains essentially constant as we always have delamination, and is proportional to the adhesive strength between the biofilm and the substrate (*Vella et al., 2009*). The adhesive strength can be readily varied in our experiments and provides a critical way to interrogate the mechanism at play, and is controllable through modification of the surface free energy of the PDMS as detailed in section Biofilm adhesion controls channel formation. The dependency of the growth of the blister on the surface free energy of the PDMS and hence the adhesion strength between the biofilm and the substrate provides evidence for the underlying delamination mechanism. The interface crack will not grow when the resistance to crack propagation, $\Gamma$, is greater than the crack driving force, $G$, and vice versa.

To summarize, the initial buckling of the biofilm is determined by the mechanical properties of the biofilm itself and the compressive stress $\sigma$ generated by the growth and volume expansion of the biofilm confined between two walls. However, the growth can simply be isotropic, in contrast to biofilms on agar, where complex differences in growth rate have been proposed to induce compressive stresses (*Ben Amar and Wu, 2014*; *Espeso et al., 2015*; *Fei et al., 2020*). Once buckled, the subsequent growth is governed by an interplay of compressive stress and interface toughness $\Gamma(\psi)$. To this end, sufficiently high stress and low adhesion of the biofilm lead to buckling delamination with the formation of a connected network of stable channels. The mechanism is quite simple and seems to rationalize the presence of such wrinkled biofilms. Additionally, the adhesive strength can be controlled by manipulating the substrate alone, which may allow us to prevent or arrest wrinkle formation without changing the material properties of the biofilms.

## Biofilm adhesion controls channel formation

Our results show that the adhesive strength between the biofilm and the substrate plays a crucial role in buckling-delamination, leading to channel formation. By tuning the interaction between the biofilm and the substrate, we can induce or impede delamination and channel formation with unprecedented control and reproducibility. The adhesion between single bacteria and a substrate can be controlled by changing the surface free energy of the substrate, as bacteria preferably adhere to surfaces with a high surface free energy (*Zhao et al., 2005*; *Callow and Fletcher, 1994*). We increase the surface free energy of PDMS from $\gamma = 23 \, \mathrm{mN \, m^{-1}}$, *Figure 4a* left panel, to $\gamma = 37 \, \mathrm{mN \, m^{-1}}$ (more hydrophilic), *Figure 4a* right panel, by adding small amounts of a PEG-PDMS block-copolymer to the PDMS mixture, following *Gökaltun et al., 2019*. Two biofilms were grown on substrates with these different surface free energies under otherwise identical conditions (*Figure 4a*). The biofilm grown on low surface free energy PDMS (left) undergoes clear buckling-delamination and develops a channel network. The biofilm grown on high surface free energy PDMS (right) does not undergo buckling-delamination and remains homogeneously adherent to the PDMS substrate. These results show that channel formation can be suppressed by increasing the surface free energy of the substrate and, consequently, the adhesion strength between the biofilm and the substrate. We vary the surface free energy of the substrate through chemical modifications (*Figure 4*) or physical modifications with oxygen plasma treatment (*Figure 4—figure supplement 1*) with identical results. We hypothesize that the overall increase in adhesion strength leads to smaller blisters where the biofilm can delaminate. According to section Buckling-Delamination as the driving force for channel formation, a smaller radius $R$ of the initial delaminated blister leads to a quadratic increase in the critical compressive stress $\sigma_c$ needed for buckling. Assuming the smallest possible delaminated blister is at the single-cell level, $\sigma_c$ becomes two orders of magnitude higher than stresses previously observed in biofilms

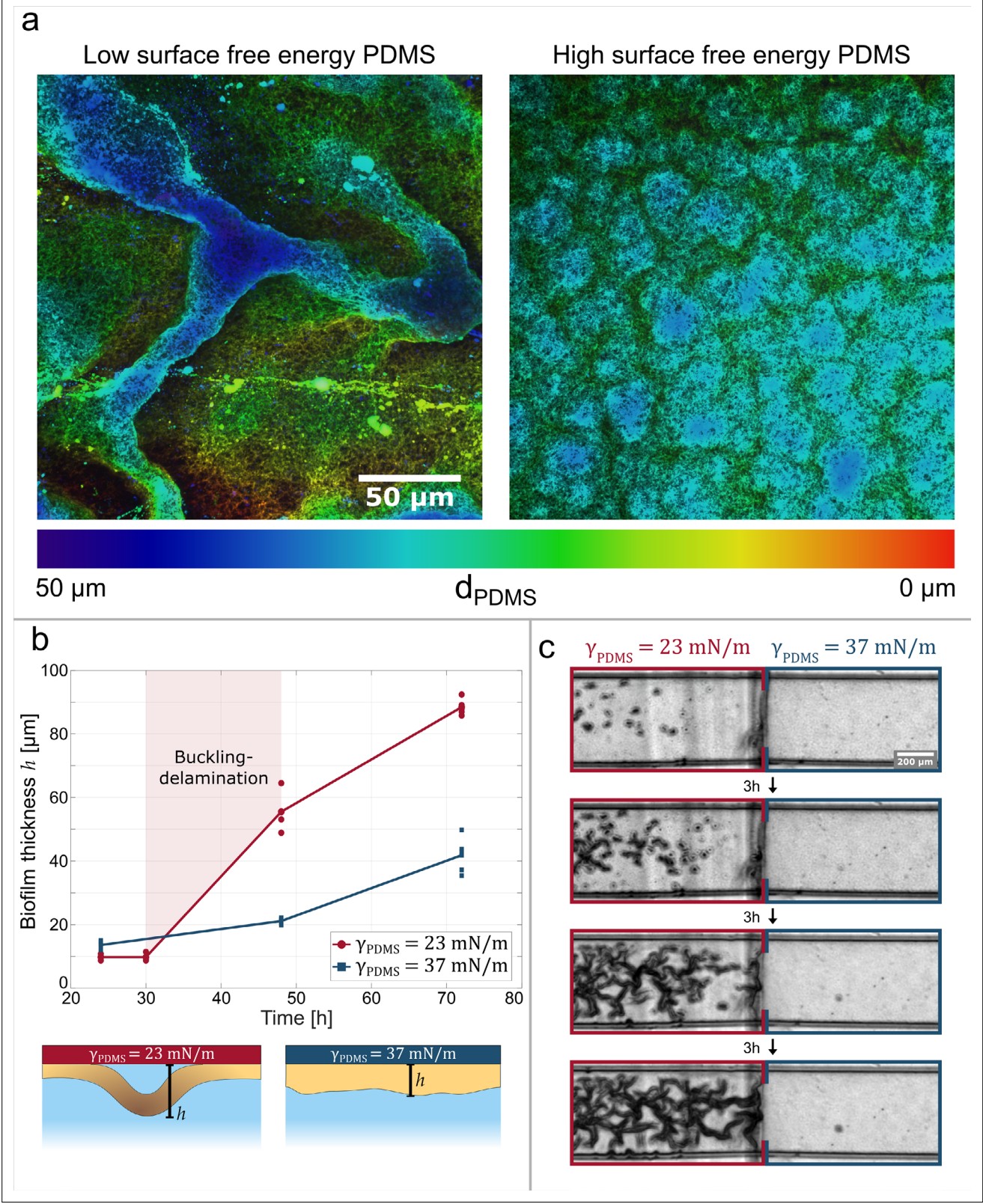

**Figure 4.** Adhesive strength between the biofilm and the substrate governs channel formation. (**a**) Reconstructions from laser-scanning confocal microscopy images of the biofilm. The biofilms are either grown on a low surface free energy PDMS substrate (left, $\gamma_{PDMS} = 23\,\mathrm{mN\,m^{-1}}$) or on a high surface free energy PDMS substrate (right, $\gamma_{PDMS} = 37\,\mathrm{mN\,m^{-1}}$). (**b**) Average effective biofilm thickness as a function of time and surface free energy. The average thickness of biofilm grown on a low surface free energy PDMS substrate (red) and on a high surface free energy PDMS substrate (blue).

*Figure 4 continued on next page*

*Figure 4 continued*

(**c**) The image sequence shows a biofilm that is grown on a patterned PDMS substrate in the same microfluidic channel. On the left, the substrate has a low surface free energy, while on the right it has a high surface free energy.

The online version of this article includes the following figure supplement(s) for figure 4:

**Figure supplement 1.** The surface free energy of the substrate can be changed through a physical process.

during growth (*Chew et al., 2016*). This suggests that in the case of strongly adhering biofilms, the compressive stress may not be sufficient to initiate buckling and supports the experimental finding that no buckling is observed in *Figure 4a*, right panel.

In the next step, we monitor the effective biofilm thickness measured from the PDMS surface to the top of the biofilm layer over time and find that, as a channel network is formed, the average biofilm thickness increases substantially and at a greater rate compared to a biofilm where no channels are formed. We obtained the effective average thickness of the biofilm by fluorescently labeling the eDNA component of the biofilm matrix with propidium iodide and measuring the thickness in the z-direction with a confocal microscope from the PDMS substrate to the top of the biofilm at the biofilm-liquid interface. *Figure 4b* compares the average, effective thickness of a biofilm grown on a high surface free energy substrate to a biofilm grown on a low surface free energy substrate. After 30–48 hr, the latter develops a channel network, and its effective thickness increases substantially. The biofilm thickness increases further with time until after 72 hr, the biofilm has reached a total thickness of roughly $90\,\mu m$ and takes up almost the whole $100\,\mu m$-high microfluidic channel. On the other hand, the biofilm grown on high surface energy PDMS does not develop a channel network. The biofilm thickness increases continuously, but slower than in the case of a channel-forming biofilm. After 72 hr, the biofilm has a mean thickness of $40\,\mu m$, less than half of the microfluidic channel height. The illustrations in *Figure 4b* clarify that the increase in effective thickness during channel formation does not necessarily correspond to an increase in growth rate or biomass compared to the non-buckled biofilm, as water-filled channels primarily constitute the additional volume.

The relation between surface free energy and buckling-delamination allows us to control the biofilm morphology depending on the substrate's surface free energy. The biofilm morphology can be controlled locally by solely adjusting the surface free energy of the substrate with a spatial resolution in the millimeter range (*Figure 4c*). We produced a microfluidic channel consisting of alternating, millimeter-long sections made of low and high-surface free energy surfaces. The biofilm grown in this patterned PDMS channel exhibits a patterned morphology that mirrors the patterning of the surface free energy of the PDMS: the biofilm grown on the low surface free energy PDMS forms a channel network, while the one grown on the high surface free energy PDMS forms a flat biofilm, with nutrient conditions being evidently equal. Remarkably, the morphological change is as abrupt as the change in surface free energy. The increase in surface free energy leads to an increase in adhesion strength and hence interface toughness ($\Gamma$). A stronger adhesion between the biofilm and the substrate leads to greater resistance against crack propagation (see section Buckling-Delamination as the driving force for channel formation). If the resistance becomes larger than the driving force, crack propagation is no longer possible, and buckling is arrested. This can be achieved by simply increasing the surface free energy of the substrate and, therefore, the adhesion strength between the biofilm and the PDMS. These experimental results show, for the first time, how basic material properties of the substrate, which moreover are easy to modulate, can be used to reliably control the biofilm morphology without changing growth conditions or biofilm composition or even enforce a patterned structure and provide a critical way to interrogate the mechanism at play.

## Bacterial movement inside the channel network

The channel network is devoid of any biofilm matrix and densely populated by actively motile bacteria, as shown by the movie of bacteria swimming in a channel *Figure 5—video 1*. The bacterial motion shows no preferential direction, and high-speed images can be used to calculate a spatially resolved time-correlation coefficient (*Secchi et al., 2013*). Bacterial motion leads to frequent local changes in the image intensity on a timescale related to the bacterial swimming speed (see section Spatially resolved degree of correlation for details). Therefore, we calculate the time and space correlation of the intensity of the image over regions of interest located in the channel and use the degree of

correlation as a representation of bacteria motility. By computing the degree of correlation of the image over time and retaining the spatial resolution, we can identify higher and lower bacterial activity areas. *Figure 5a* shows the activity maps and the corresponding brightfield images at different stages of the biofilm development. The first stage corresponds to a time of 72 hr after the start of the experiment and shows the biofilm, roughly one hour before it starts to form a channel network. The activity map shows a uniformly high degree of correlation and, therefore, no detectable bacteria movement. Nine hours later, the channel network is fully developed according to the bright-field micrograph, and the activity map shows large, active areas with a low degree of correlation. It becomes clear that areas with detectable bacteria movement are highly localized and distinct from inactive areas. A comparison between the activity map and the biofilm microstructure, as shown in the bright-field micrographs, reveals that the active areas are exclusively found inside the channels of the biofilm. These results indicate that the hollow channel network gets populated by motile bacteria as the channels form.

As the biofilm matures, the bacterial activity diminishes until it can not be detected anymore (*Figure 5a*, right panel), without any structural changes in the biofilm. Previous studies on biofilm dispersal have described a mode of dispersal known as seeding dispersal, where a large number of single bacteria are released from hollow cavities formed inside the biofilm colony (*Kaplan, 2010*). In non-mucoid PAO1 biofilms, these hollow cavities are filled with motile, planktonic cells before a breach in the biofilm wall releases the cells into the surrounding medium (*Purevdorj-Gage et al., 2005*). In our case, we observe that the channels get filled with planktonic cells. However, we do not observe any dispersal; the channel walls were not eroded and remain intact. The right panel in *Figure 5a* shows the same section of the biofilm two hours after the maximum movement inside the channels is detected. The activity map shows that the previously active areas have changed into areas with a high degree of correlation and hence no detectable bacterial activity. The corresponding phase-contrast image reveals that the decrease in bacterial motility comes without a dispersal event nor deformation or structural changes of the biofilm. We hypothesize that this decrease in bacterial motility is attributable to a renewed surface colonization and subsequent biofilm formation of the planktonic cells in the channels.

The swimming speed of the motile bacteria inside the channel network is not affected by the fluid flow surrounding the biofilm. Since previous studies suggested that channels in biofilm introduce flow to overcome diffusion-limited transport of nutrients (*Wilking et al., 2012*), we verified if the nutrient flow could induce advective transport inside the channel network. To this end, we performed differential dynamic microscopy (DDM) to extract the average bacterial swimming speed of the bacteria inside the channels (*Bayles et al., 2016*; *Wilson et al., 2011*). The details of DDM are explained by *Cerbino and Trappe, 2008* and are summarized in the methods-section Differential Dynamic Microscopy. The average bacterial swimming speed was measured as a function of the mean flow rates of the nutrient solution surrounding the biofilm. The results in *Figure 5c* show no clear dependency of the average bacterial swimming speed inside the channels from the flow velocity of the nutrient solution, despite the flow velocity of nutrients varying from $0\,mm\,s^{-1}$ to $11.1\,mm\,s^{-1}$ and being three orders of magnitude larger than the bacterial swimming speed. In addition, the average value of the swimming speed (20 to $30\,\mu m\,s^{-1}$) corresponds to values previously reported in the literature for *P. aeruginosa* PAO1 in suspension (*Khong et al., 2021*). These findings indicate that the bulk flow surrounding the biofilm does not introduce advection inside the biofilm, and therefore the channels consist of a closed biofilm matrix layer populated by motile bacteria.

## Discussion

We reported for the first time the structural evolution of biofilm grown on a solid substrate exposed to fluid flow in a microfluidic device. A buckling-delamination process governs the formation of three-dimensional hollow channels. Experimentally, we show that the biofilm morphology is determined by the isotropic growth of the biofilm in a confined space and the adhesion between the biofilm and the solid substrate. These findings give unprecedented control over the biofilm morphology through basic physical parameters such as adhesive strength to the substrate and nutrient concentration.

Our results show that biofilm growth is the key driving force for buckling-delamination that leads to the formation of channels. The continuous growth of a biofilm in a confined space induces compressive stresses that initiate buckling of the biofilm. Previous studies have identified growth-induced compressive stresses to play a role in the wrinkling of biofilm grown on agar plates (*Asally*

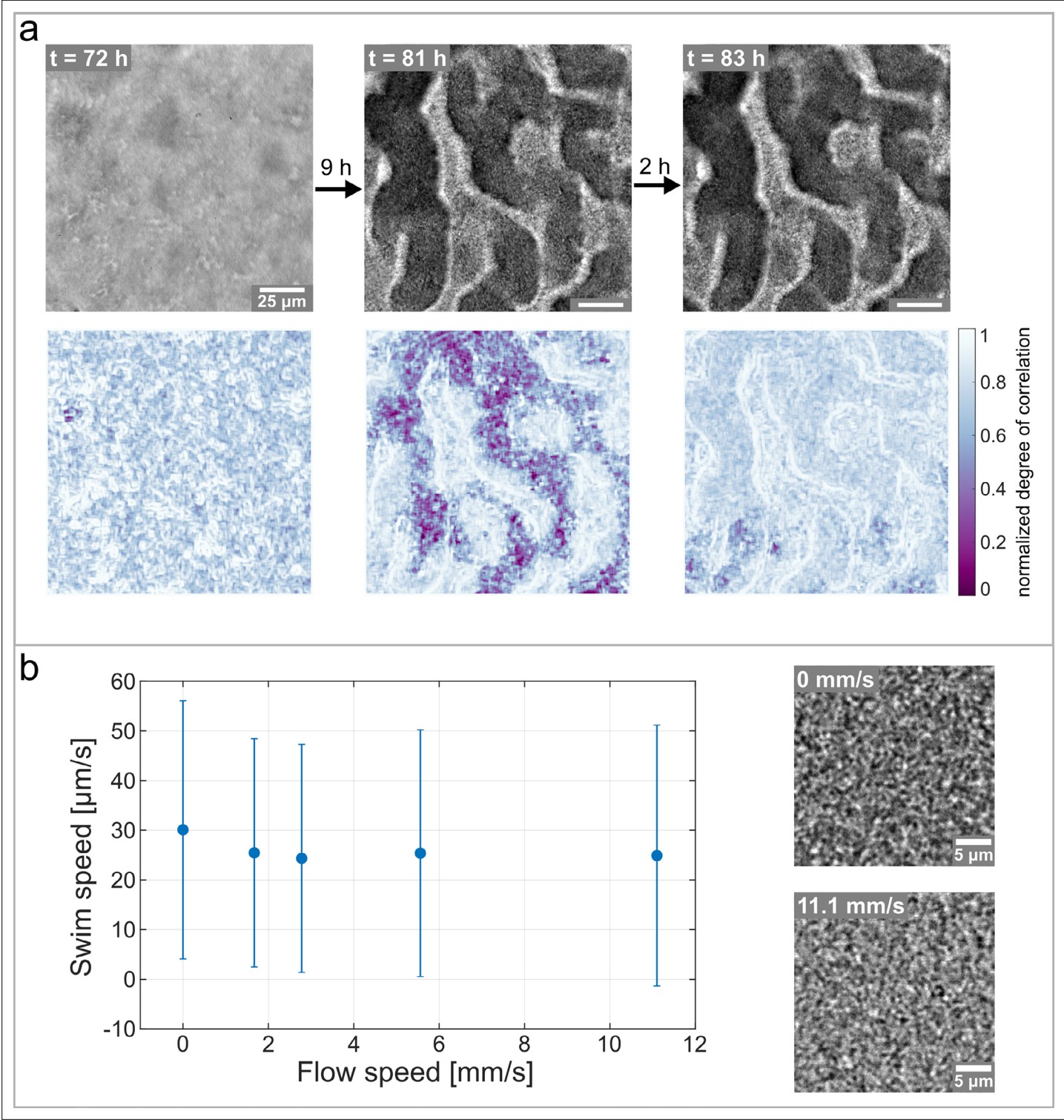

**Figure 5.** Hollow channels are populated by motile bacteria. (**a**) Bacterial movement inside the channel network visualized using a spatially resolved, normalized degree of correlation. A low degree of correlation corresponds to an active region. The brightfield images show the corresponding structure of the biofilm. (**b**) Differential Dynamic Microscopy is used to quantify the bacterial swimming speed inside the biofilm channels. The swimming speed is measured at varying fluid flow speeds inside the microfluidic device. The two microscopy images show a close up of the bacterial biofilm at two different fluid flow velocities. The errorbars indicate the standard deviation from the mean swim speed. The videos of the bacteria motion can be found in *Figure 5—video 2* and *Figure 5—video 3*.

The online version of this article includes the following video for figure 5:

*Figure 5 continued*

**Figure 5—video 1.** Video of the bacterial movement inside the biofilm channels.
https://elifesciences.org/articles/76027/figures#fig5video1

**Figure 5—video 2.** Video of bacteria movement with no surrounding fluid flow.
https://elifesciences.org/articles/76027/figures#fig5video2

**Figure 5—video 3.** Video of bacteria movement with a surrounding fluid flow velocity of .
https://elifesciences.org/articles/76027/figures#fig5video3

*et al., 2012*; *Yan et al., 2019*; *Ben Amar and Wu, 2014*; *Espeso et al., 2015*; *Fei et al., 2020*). In these systems, the diffusion-limited transport of nutrients exclusively from the bottom of the biofilm can lead to gradients in growth rate. In combination with weak adhesion to the agar, the spatial differences in growth rate may induce compressive stresses that initiate wrinkling. In our case, the introduction of moderate fluid flow increases nutrient flux at the surface of the biofilm and therefore minimizes nutrient gradients in the bulk of thin biofilms (*Krsmanovic et al., 2021*). Therefore, we can assume a uniform biofilm in x-y direction and minimal growth gradients in the z-direction, even though local heterogeneities in growth can still occur. These findings also show that in the simple system of confined growth of a uniform biofilm, compressive stresses are high enough to induce buckling and channel formation, a result that has recently been seen in bacterial pellicles by *Qin et al., 2021*.

This work emphasizes the importance of mechanical instabilities in biofilm wrinkling and elucidates the dependence of the channel formation process on the adhesive strength between the biofilm and the solid substrate. In our system, the biofilm delaminates and buckles away from the substrate to form a channel network. Previous studies found that wrinkled biofilms often exhibit a layered structure where the top layer wrinkles and the bottom layer stays bonded to the agar plate (*Yan et al., 2019*; *Zhang et al., 2016a*; *Zhang et al., 2017*). However, we observe biofilm delamination directly from the substrate without any intermediate layer. This further confirms that our experimental setup leads to the formation of homogeneous, non-layered biofilms. Additionally, we showed experimentally that an increase in adhesive strength between the biofilm and the substrate impedes channel formation, as the biofilm can no longer delaminate. This understanding gives us full control over biofilm morphology: we patterned and predicted the biofilm structure based on the surface free energy of the PDMS substrate.

Many recent studies focused on static biofilm-agar systems to describe and understand the mechanical contributions to the structural evolution of biofilms. However, nutrient gradients, spreading and swarming of colonies, or the mechanical response of the substrate complicate the analysis and may convolute the purely mechanical contributions with biological responses of the microorganisms. We show that, within well-defined microfluidic systems, it is possible to isolate the mechanical contributions from the biofilm structure and control them without changing any biological parameter. Furthermore, we hypothesize that our findings are general and applicable to other bacterial species as our growth conditions - fluid flow and the presence of solid substrates – are often found in the biofilms habitats. This might open up new strategies for biofilm control and contribute to a more holistic view of biofilm formation and evolution.

## Materials and methods

### Culture conditions and growth in the microfluidic device

*Pseudomonas aeruginosa* PAO1 wild-type (WT) was grown in tryptone broth ($10\,g\,l^{-1}$ Tryptone, microbiologically tested, Sigma Aldrich, $5\,g\,l^{-1}$ NaCl) in an orbital shaker overnight at 37. The overnight culture was then diluted 1:100 in tryptone broth (TB) and grown for 2 hr until $OD_{600}$ reached the value of 0.2. The bacterial suspension was then diluted 1:10 and used to inoculate the microfluidic channel.

The microfluidic channel was inoculated by withdrawing $600\,\mu l$ of bacterial suspension from a 2 mL Eppendorf tube. The bacteria were left undisturbed for 1 hr before fresh media was flown using the syringe pump. For all microfluidic experiments, a diluted 1:10 tryptone broth ($1\,g\,l^{-1}$ Tryptone, $5\,g\,l^{-1}$ NaCl) was used as the growth medium and the temperature was kept constant at $25\,^{\circ}C$.

## Microfluidic device

Rectangular microfluidic channels were fabricated using standard soft lithography techniques (*Xia and Whitesides, 1998*). First, microchannel molds were prepared by depositing SU-8 2,150 (Micro-Chem Corp., Newton, MA) on silicon wafers via photolithography. Next, polydimethylsiloxane (PDMS; Sylgard 184 Silicone Elastomer Kit, Dow Corning, Midland, MI) was prepared and cast on the molds. After curing for 24 hr at $80\,°C$, PDMS microchannels were plasma-sealed onto a clean glass slide. The PDMS channels were flushed with $2\,ml$ of fresh media before each experiment. Flow was driven by a syringe pump (Standard PHD Ultra syringe pump, Harvard Aparatus), and the flow velociy was held constant at $1.7\,mm\,s^{-1}$ during the experiment.

Hydrophilic PDMS with 0.5% dimethylsiloxan-ethyleneoxide blockcopolymer (DBE-712, Gelest, Morrisville, PA) was produced according to *Gökaltun et al., 2019*. Casting and plasma bonding were carried out as described above. The patterned microfluidic channel was produced by first casting hydrophobic PDMS onto the molds and curing the PDMS as described above. Then, millimeter-long sections were cut out with a precision blade without removing the PDMS from the mold. The removed sections were filled with hydrophilic PDMS. Finally, the patterned channels were cured and bonded to a glass slide as described above.

The surface free energies of the hydrophilic and hydrophic PDMS were determined with the Owens-Wendt-Method (*Owens and Wendt, 1969*), where the contact angles of known liquids with the substrate are used to determine the unknown surface free energy of the substrate. To that extent, the contact angles of Nitromethane ($\gamma_L =36.8\,\mathrm{mN\,m^{-1}}$, *Jańczuk and Białłopiotrowicz, 1989*) Hexadecane ($\gamma_L =26.35\,\mathrm{mN\,m^{-1}}$, *Jańczuk et al., 1993*) and Water ($\gamma_L =72.8\,\mathrm{mN\,m^{-1}}$, *Zhang et al., 2019*) on hydrophilic and hydrophobic PDMS were measured.

## Staining procedures

Staining with a propidium iodide solution was performed to measure the thickness of the biofilm. We produced the staining solution by mixing propidium iodide (Sigma Aldrich) with the nutrient medium to a final concentration of $5\,\mathrm{M}$ and flowed the solution for the entire experiment duration. GFP-labelled Concanavalin A (Sigma Aldrich) was used to visualize the three-dimensional structure. The stain was dissolved in the nutrient solution to a final concentration of $100\,\mathrm{g\,l^{-1}}$. The biofilm was incubated for 20 min with the staining solution before being washed with a fresh nutrient solution.

## Visualization

Light microscopy images were taken on Nikon Eclipse Ti2-A in phase-contrast configuration, equipped with a Hamamatsu ImageEM-X2 CCD camera and a 20 x objective. For timelapse images, we used the microscope control software μManager (*Stuurman et al., 2007*) and acquired an image every 5 min. The phase-contrast images were analysed with the software Fiji (*Schindelin et al., 2012*). Fiji was also used to produce the three-dimensional renderings of the biofilm from the confocal images using the temporal color code function. For the fluorescent visualizations, we used a Nikon Eclipes T1 inverted microscope coupled with a Yokogawa CSU-W1-T2 confocal scanner unit and equipped with an Andor iXon Ultra EMCCD camera. The images were acquired with a 60 x water immersion objective with N.A. of 1.20. We used Imaris (Bitplane) for analysing and producing cross-sections of the z-stacks.

## Skeletonization of channel networks

The quantitative analysis of the channel network formation was performed using Fiji and Matlab (version 9.7.0 (R2019b)). Natick, Massachusetts: The MathWorks Inc, 2019. As a first step, the bright-field timelapse images were binarized with Fiji. Otsu's method (*Otsu, 1979*) was used to determine the thresholding value of the last image of the timelapse, and this thresholding value was used to binarize all images. Next, the binarized images were imported into Matlab and objects smaller than 5 pixels were removed and a morphological opening operation was performed with Matlabs own function *bwareaopen* before the resulting images were skeletonized using Matlab skeletonization command *bwskel*. Finally, the Matlab function *bwlabel* was used to label all connected components of the skeletonized image and extract the longest connected path and the total number of wrinkles.

## Differential dynamic microscopy

Images were acquired at 2000 frames per second with the Fastcam UX100 (Photron, Japan) high-speed camera on the Nikon Eclipse Ti2-A microscope in brightfield mode. Differential Dynamic Microscopy (DDM) was performed according to *Cerbino and Trappe, 2008* using a custom code written in Matlab. Subsequent fitting and bacterial swimming speed extraction was performed as described by *Wilson et al., 2011*. The theory of DDM is described in detail by *Cerbino and Trappe, 2008*. In short, we calculate the difference between images of our time-lapse recording of the motile bacteria at different time intervals. From the Fourier transform of the image difference, we obtain the intensity correlation function related to the bacterial local dynamics. Analyzing and fitting the correlation function allows us to extract the diffusive and active contributions of the bacterial movement and hence gives us a good estimation of the swimming speed of bacteria.

## Spatially resolved degree of correlation

Images were acquired at 1000 frames per second with the Fastcam UX100 (Photron, Japan) high-speed camera on the Nikon Eclipse Ti2-A microscope in brightfield mode. The spatially resolved correlation coefficient $c_I(\tau; t, r)$ between two images taken at times $t$ and $t + \tau$ was calculated according to *Secchi et al., 2013*

$$c_I(\tau; t, r) = \frac{\langle I_p(t)I_p(t+\tau)\rangle_r}{\langle I_p(t)\rangle_r \langle I_p(t+\tau)\rangle_r} - 1. \tag{3}$$

$I_p$ is the image intensity measured by the $p^{th}$ pixel and $\langle...\rangle_r$ denotes an average over all pixels within a region of interest centered around $r$. The images were subdivided into regions of interest of $2.5 \times 2.5 \, \mu m$. The degree of space-time correlation was calculated between two images which were $1\,s$ apart and averaged over the regions of interest. This correlation coefficient was calculated for 200 images with the same timestep and averaged.

## Acknowledgements

The authors acknowledge Matteo Brizzioli, Giovanni Savorana and Dr. Alexandra Bayles for their contributions to the DDM code and analysis; Dr. Kirill "Hipster" Feldman and Ela Burmeister for their valuable experimental support and the ScopeM facility at ETH Zurich for providing excellent equipment and support. E S acknowledges support from SNSF PRIMA grant 179,834.

## Additional information

### Funding

| Funder | Grant reference number | Author |
|---|---|---|
| Schweizerischer Nationalfonds zur Förderung der Wissenschaftlichen Forschung | 179834 | Eleonora Secchi |

The funders had no role in study design, data collection and interpretation, or the decision to submit the work for publication.

### Author contributions

Steffen Geisel, Conceptualization, Investigation, Methodology, Visualization, Writing - original draft, Writing - review and editing; Eleonora Secchi, Investigation, Methodology, Supervision, Writing - original draft, Writing - review and editing; Jan Vermant, Conceptualization, Funding acquisition, Supervision, Writing - original draft, Writing - review and editing

### Author ORCIDs

Steffen Geisel http://orcid.org/0000-0002-1121-3103
Eleonora Secchi http://orcid.org/0000-0002-0949-9085
Jan Vermant http://orcid.org/0000-0002-0352-0656

**Decision letter and Author response**
Decision letter https://doi.org/10.7554/eLife.76027.sa1
Author response https://doi.org/10.7554/eLife.76027.sa2

## Additional files

### Supplementary files
• Transparent reporting form

### Data availability
The raw data of the graphs in Figures 1, 3, 4 and 5 are made available through a Dryad repository as timelapse and High-Speed images. The Matlab Codes for Skeletonization (Figure 1), DDM and Correlation calculations (Figure 5) are made available in the same repository. The link to the repository can be found in the Materials and Methods section or under: https://doi.org/10.5061/dryad. vq83bk3tn.

The following dataset was generated:

| Author(s) | Year | Dataset title | Dataset URL | Database and Identifier |
|---|---|---|---|---|
| Geisel S, Secchi E, Vermant J | 2022 | Data from: The role of surface adhesion on the macroscopic wrinkling of biofilms | https://doi.org/ 10.5061/dryad. vq83bk3tn | Dryad Digital Repository, 10.5061/dryad.vq83bk3tn |

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
