## [Editor Report]

The wrinkling of growing biofilms is considered in this paper experimentally in a clever set of experiments in a microfluidic setup that reveals aspects of the onset of the wrinkling instability and the formation of hollow channels within which bacteria move. Variations in the adhesive properties of the underlying surface are shown to affect the instability.

---

## [Decision Letter]

**Decision letter after peer review:**

Thank you for submitting your article "The role of surface adhesion on the macroscopic wrinkling of biofilms" for consideration by *eLife*. Your article has been reviewed by 2 peer reviewers, one of whom is a member of our Board of Reviewing Editors, and the evaluation has been overseen by Aleksandra Walczak as the Senior Editor. The reviewers have opted to remain anonymous. We regret the lengthy delay in furnishing this decision letter.

Essential revisions:

1. The main shortcoming of this paper is that the theory is disconnected from the experimental findings. Most of the theory is extracted directly from the cited paper by Hutchinson et al., (1992), but with little or no explanation. Moreover, none of the quantities given by Equations 1-3 are measured or estimated in the experiments, and therefore are not used (or useful) to rationalize the experimental observations. Some parameters in the equations are not even defined: specifically interface toughness and ψ are nowhere defined in the manuscript. Because of this issue, it is not clear at all how the adhesion energy enters into the theory nor how it can be used for instance to explain the results shown in Figure 4c, where under the same compression stress, the biofilm only transitions to a wrinkled state in the case of lower surface energy. Related to these issues, it is also not clear what sets the initial radius R of the buckled spots of the biofilm observed in Figure 1b. This seems like an interesting and important question to address and is definitely not addressed by the theory as presented which simply assumes R as a parameter. All these issues need improvement and clarification if the authors want to include a relevant theoretical section.

2. Several other points are not adequately explained. The authors should at least briefly explain the method by which they measure the swimming speed of bacteria inside the channels. Similarly, how is the surface energy measured in both cases?

---

## [Author Response]

Essential revisions:1. The main shortcoming of this paper is that the theory is disconnected from the experimental findings. Most of the theory is extracted directly from the cited paper by Hutchinson et al., (1992), but with little or no explanation. Moreover, none of the quantities given by Equations 1-3 are measured or estimated in the experiments, and therefore are not used (or useful) to rationalize the experimental observations. Some parameters in the equations are not even defined: specifically interface toughness and ψ are nowhere defined in the manuscript. Because of this issue, it is not clear at all how the adhesion energy enters into the theory nor how it can be used for instance to explain the results shown in Figure 4c, where under the same compression stress, the biofilm only transitions to a wrinkled state in the case of lower surface energy. Related to these issues, it is also not clear what sets the initial radius R of the buckled spots of the biofilm observed in Figure 1b. This seems like an interesting and important question to address and is definitely not addressed by the theory as presented which simply assumes R as a parameter. All these issues need improvement and clarification if the authors want to include a relevant theoretical section.

We thank the Editor and the Reviewers for pointing this out. We have revised the section “Buckling-Delamination as the driving force for channel formation”, starting on line 172, and tried to connect the theory and the experimental findings. In particular,

– We estimated the critical compressive stress needed for the biofilm to buckle away from the substrate using Equation 1 in the typical experimental conditions. The estimated value of the critical stress corresponds favorably to compressive stresses observed in *P. aeruginosa* biofilms during growth (Chew 2016), supporting the hypothesis that a buckling-delamination process controls channel formation. We added estimates of the critical stress based on literature value and discussed the growth of a blister on page 7 (from line 204 onwards).

– We added discussion on how the experimental observations in Figure 4 b and c, in particular, relate to the theoretical findings, both when it comes to the increase in effective thickness due to the phenomenon of buckling as well as the effects of the changes in surface free energy on the adhesion strength and how this suppresses interface crack propagation. We have clarified and extended the text (lines 264-279, lines 287-290, and lines 300-310).

– We verified that all parameters are now defined.

2. Several other points are not adequately explained. The authors should at least briefly explain the method by which they measure the swimming speed of bacteria inside the channels. Similarly, how is the surface energy measured in both cases?

We added a brief explanation of the theory of differential dynamic microscopy (DDM) in the Materials and methods section (lines 475-481) and referred the reader to the work of Cerbino and Trappe for a detailed explanation of the technique. We clarify the application to the characterization of bacterial motility, as this could be useful for a broader community.

We also added the specific procedure of contact angle measurement, which was used to measure the surface free energy of the PDMS substrates (lines 435-440). The procedure is a standard textbook method, and we compare the literature values to clarify this point further.

As we emphasize that the surface energy plays a role in the force balance, we prefer to keep the units as a force per unit length.